# Co-Delivery of Methotrexate and Nanohydroxyapatite with Polyethylene Glycol Polymers for Chemotherapy of Osteosarcoma

**DOI:** 10.3390/mi14040757

**Published:** 2023-03-29

**Authors:** Lingbin Ou, Qiongyu Zhang, Yong Chang, Ning Xia

**Affiliations:** 1School of Medical Technology, Yongzhou Vocational Technical College, Yongzhou 425100, China; 2College of Chemistry and Chemical Engineering, Anyang Normal University, Anyang 455000, China

**Keywords:** osteosarcoma, methotrexate, nanohydroxyapatite, polyethylene glycol, drug delivery

## Abstract

Neoadjuvant chemotherapy is an alternative treatment modality for tumors. Methotrexate (MTX) has been often used as a neoadjuvant chemotherapy reagent for osteosarcoma surgery. However, the large dosage, high toxicity, strong drug resistance, and poor improvement of bone erosion restricted the utilization of methotrexate. Here, we developed a targeted drug delivery system using nanosized hydroxyapatite particles (nHA) as the cores. MTX was conjugated to polyethylene glycol (PEG) through the pH-sensitive ester linkage and acted as both the folate receptor-targeting ligand and the anti-cancer drug due to the similarity to the structure of folic acid. Meanwhile, nHA could increase the concentration of calcium ions after being uptake by cells, thus inducing mitochondrial apoptosis and improving the efficacy of medical treatment. In vitro drug release studies of MTX-PEG-nHA in phosphate buffered saline at different pH values (5, 6.4 and 7.4) indicated that the system showed a pH-dependent release feature because of the dissolution of ester bonds and nHA under acidic conditions. Furthermore, the treatment on osteosarcoma cells (143B, MG63, and HOS) by using MTX-PEG-nHA was demonstrated to exhibit higher therapeutic efficacy. Therefore, the developed platform possesses the great potential for osteosarcoma therapy.

## 1. Introduction

Osteosarcoma is the most frequent type of primary malignant bone tumor. It mainly occurs in children, adolescents, and young adults, and it can transfer from bones to other parts (e.g., lungs, breast, and prostate) [1]. After adequate clinical treatment, the five-year overall survival for osteosarcoma patients is still less than 60% [2]. In the initial stage, surgery is the standard treatment for osteosarcoma, but this seriously endangers the physical and mental health of patients. At present, chemotherapy is an important and feasible clinical treatment strategy for suppressing the progression of osteosarcoma [3]. As a frequently used chemotherapeutic drug, methotrexate (MTX), with a folic acid-like structure, has become the backbone of the osteosarcoma therapy since 1970s [4]. It not only can be internalized by folate receptors-overexpressed cancer cells through a folate-like transport mechanism, but it also can interact with dihydrofolate reductase, thereby inhibiting the metabolism of folic acid and suppressing the synthesis of NDA/RNA [5,6]. However, the poor water solubility and pharmacokinetic property and low bioavailability of MTX dramatically limited its clinical efficacy. Furthermore, the high-dose MTX pulse therapy always causes drug resistance by target cells and serious adverse side effects, including immune suppression, myelosuppression, hepatotoxicity, and cardiotoxicity [7,8]. Thus, it is of great importance to develop appropriate means for efficient and systemic drug delivery to improve the drug efficacy with decreaseddosage.

During the past decades, various nanomaterials have been successfully introduced into the development of versatile nanoscaled drug delivery systems to reduce side effects and to address the drug resistance issue [9]. Benefitting from the chemical/compositional similarity to the inorganic phase of skeletal bones and teeth in animals, hydroxyapatite (HA), with the structural formula of (Ca_10_(PO_4_)_6_(OH)_2_), exhibits adequate biodegradation, excellent biocompatibility, and high osteoconductivity. HA has been extensively explored in the application of orthopedic and dental implants in recent decades. Meanwhile, nanosized hydroxyapatite particles (nHA) have been widely used to act as nanocarriers for DNA, proteins, antibiotics, and anticancer drugs [10,11,12,13]. Besides acting as the delivery vehicle, nHA has been demonstrated to inhibit the proliferation of several types of tumor cells, such as osteosarcoma, gastric cancer, colon cancer, and breast cancer [14,15]. Previous studies have shown that nHA exhibited higher bioactivity than macroscale HA [16]. It can be closely combined with human tissue in a short time after surgery and effectively repair the damaged bone in the bone defect area. More importantly, it was reported that nHA can move through the cell membrane into the cytoplasm via an endocytic pathway, eventually inhibiting the growth of osteosarcoma cells [16,17]. nHA can also induce mitochondrial apoptosis by destroying mitochondrial membrane and leading to the release of Cyt C, that is, nHA has a certain anticancer effect [18,19]. Besides, nHA combined with polymers and drugs can be utilized as high-strength composite artificial bone for a strategic local drug delivery [20]. The traditional drug-loaded artificial bone is implanted by surgery, making it difficult to adjust drug dosage flexibly. Therefore, it is a promising approach to construct biocompatibility and pH-responsiveness as they relate to nHA-based targeted nanocarriers to overcome shortcomings.

Generally, almost all the reported nanosized drug delivery systems rely on passive or active tumor-targeting abilities [21]. Passive targeting mainly depends on the characteristic features of tumor issues, including the leaky walls and poor lymphatic drainage, to prolong circulation times in vivo and accumulate at particular sites via the enhanced permeability and retention effect (EPR) for relatively large-sized nanoparticles (15–400 nm). However, the inherent biophysicochemical properties of nanomaterials may limit the effective concentration of NPs at active sites because of the competitive sequestration by the cells of the mononuclear phagocytic system. Conversely, active targeting can be achieved by the conjugation of affinity ligands or biomolecules on the surface of nanomaterials containing chemotherapeutics [22,23]. After the binding between ligands and receptors over-expressed on the surface of target cells, the actively targeted nanomaterials could be internalized through different receptor-mediated endocytosis pathways, thus enhancing the therapeutic efficacy significantly. For example, the folate receptor is frequently over-expressed in a vast array of tumor cells, and many folate-modified nanomaterials have been utilized to deliver drugs to folate receptor-positive tumors [24,25,26]. Thanks to the structure similarity to folate, MTX as the anticancer drug can also be employed as the active targeting ligand to self-direct MTX-conjugated drug delivery vehicles to specifically enter into tumor cells overexpressing folate receptors [27].

In this work, we developed a nHA-based MTX delivery system with cancer cell self-targeting and treatment ability for in vitro characterization and investigation of osteosarcoma treatment (Figure 1). nHA can function as a MTX-carrying scaffold and inhibit cell growth. MTX was conjugated to the polymer on the surface of nHA through the PEG linker, which acted as not only the drug, but also the targeting ligand toward folate receptors, which are over-expressed on the surface of cancer cells. In addition, PEG could improve the solubility and therapeutic efficiency of drugs and enhance the stability of nanocomposites and the efficiency of the particle internalization by target cells. After the internalization into cells, nHA would be degraded in the acidic environment of endosome and endolysosomes. In this process, the ester bond formed by the hydroxyl group of nHA and the carboxyl group of PEG would be hydrolyzed, leading to the release of carried MTX drugs.

## 2. Materials and Methods

### 2.1. Materials

MTX, NH_2_-polyethylene glycol-COOH (NH_2_-PEG-COOH, Mw ~ 2000), N,N’-carbonyldiimidazole (CDI), and nHA were purchased from Macklin Inc. (Shanghai, China). 1-(3-(Dimethylamino)propyl)-3-ethylcarbodiimide (EDC), dimethyl sulfoxide (DMSO), and 4-dimethylamino-pyridine (DMAP) were supplied by Sigma-Aldrich (Shanghai, China).

### 2.2. Instruments

UV-visible spectra were collected using a UV-visible spectrophotometer (Shimadzu, UV-1900i, Kyoto, Japan). The Fourier transform infrared (FTIR) absorption spectra of different samples were collected by an IR spectrophotometer (Thermo Fisher Nicolet iS5 FTIR, Waltham, MA, USA) in the wave number region of 400 to 4000 cm^−1^ using KBr pellets. ^1^H NMR spectra were recorded using a Bruker instrument (Bruker AV-500, Billerica, MA, USA). The particle size, surface potential, and polydispersity index were determined using Malvern Mastersizer S90 (Malvern Co, Worcestershire, UK). A transmission electron microscope (TEM) image was obtained on a Tecnai G2 F20 system (FEI Co., Hillsboro, OR, USA) operating at an accelerating voltage of 200 kV. Cell morphology images were recorded by fluorescence microscope (Motic Co., Xiamen, China).

### 2.3. Synthesis of MTX-PEG-COOH

MTX-PEG-COOH was synthesized through a two-step process, according to the previously reported work with slight modification [28]. 0.2 g of MTX, 0.1 g of EDC and 0.05 g of DMAP were dissolved in 5 mL of anhydrous DMSO and stirred at 40 °C for 1 h to activate the carboxyl group in MTX. Then, 0.4 g of NH_2_-PEG-COOH was added to the active ester of MTX and the solution was stirred for 24 h at room temperature. After that, the solution was poured into a dialysis bag and dialyzed against water for two days. The resulting liquid was lyophilized to obtain MTX-PEG-COOH power.

### 2.4. Synthesis of MTX-PEG-nHA

A typical synthesis of MTX-PEG-nHA was described as follows [29]: 0.1 g of nHA powder and 0.1 g of CDI were dispersed in 10 mL DMSO under ultrasonication for 1 h in ice bath. An amount of 0.1 g of the synthesized MTX-PEG-COOH and 0.02 g of DMAP were dissolved in 5 mL DMSO under continuous stirring at 40 °C for 1 h. Then, the two solutions were mixed and sonicated for 1 h. After that, the mixture was kept in an oil bath for one day. The suspension was centrifuged, and the solid was washed with water three times. After the freeze dry, MTX-PEG-nHA power was collected.

### 2.5. Characterization

Generally, 5 mg of the as-synthesized MTX-PEG-nHA powder was added into 5 mL of phosphate buffered saline solution (PBS) solution and treated with ultrasonication for 3 min to obtain MTX-PEG-nHA solution. For TEM analysis, several droplets were mounted on copper grids, followed by sample drying at room temperature. An amount of 1 mL of MTX-PEG-nHA solution was added into the sample cell for particle size and surface potential analysis, respectively.

### 2.6. Drug-Loading Capacity and In Vitro Drug Release

Firstly, the MTX content in MTX-PEG-COOH and the loading capacity of MTX-PEG-nHA were investigated. The absorption intensity of MTX solution at different concentrations (0, 2, 4, 6, 8 and 10 μg/mL) was measured by UV-vis spectrophotometry at 302 nm. A calibration curve was established using the absorption intensity of MTX solution at various concentrations. The absorbance of MTX-PEG-COOH solution (20 μg/mL) and MTX-PEG-nHA solution (30 μg/mL) were then measured at 305 nm, and the concentrations were calculated based on the standard curve, respectively. The drug loading efficiency was calculated according to the following formula:$$\mathrm{Drug}\;\mathrm{loading}\;\mathrm{efficiency}=\frac{\mathrm{weight}\;\mathrm{of}\;\mathrm{loaded}\;\mathrm{drug}}{\mathrm{weight}\;\mathrm{of}\;\mathrm{drug}\;\mathrm{in}\;\mathrm{feed}}\;\times \;100$$

To evaluate the drug release profiles, 5 mL of MTX-PEG-nHA solution was wrapped on dialysis bags (molecular weight cut-off 7000 Da) in 50 mL of PBS solution with three different pH values (5, 6.4 and 7.4) and incubated at 37 °C with shaking at 50 rpm. The supernatant was withdrawn at predetermined time intervals (0, 1, 2, 4, 8, 12, 24 and 48 h) for all the batches, and 50 mL of fresh PBS was added. The amount of the drugs released into the media was calculated by measuring the absorbance of samples at 302 nm and employing the calibration curve of MTX in PBS with a given pH value. The cumulative percent drug release (*E*_r_) was evaluated using the following equation:$$E_{r}\;(\%)=\frac{\text{∑}c_{t}V_{t}}{c_{0}V_{0}}\;\times \;100$$

### 2.7. Cell Culture

Human skin fibroblasts (HSFs) and human osteosarcoma cell line (143B, MG63 and HOS)were routinely cultured in Dulbecco’s modified Eagle’s medium (DMEM) containing 10% fetal calf serum (FBS) and 1% penicillin-streptomycin in a humidified incubator containing 5% CO_2_ at 37 °C. Every two days, the cells were digested with EDTA-free trypsin, and the medium was changed. After the seventh generation, the cells were used for further cell testing.

### 2.8. Measurement of Cell Viability

A standard CCK-8 assay was used to assess the cell viability after treatment with free MTX, MTX-PEG-COOH, and as-synthesized MTX-PEG-nHA, respectively. Briefly, different cells were placed in a 96-well cell culture plate at a density of 5000 cells per well and incubated at 37 °C overnight. Then, the medium was replaced with PBS solution, containing corresponding samples at a fixed concentration. After that, the medium was removed, and each experimental well was carefully washed with PBS solution three times. Next, CCK-8 solution was added to each well and incubated for 4 h. The absorbance was measured at 450 nm using a micro-plate reader. Cell viability was calculated using the following formula:$$\mathrm{Cell}\;\mathrm{viability}\;\left(\%\right)=\frac{{\mathrm{OD}}_{\mathrm{test}}-{\mathrm{OD}}_{\mathrm{blank}}}{{\mathrm{OD}}_{\mathrm{control}}-{\mathrm{OD}}_{\mathrm{blank}}}\;\times \;100$$
where OD_test_ and OD_control_ represent the optical intensity measured at 450 nm for treated cells and control cells, respectively, and OD_blank_ is the intensity of the wells without cells.

## 3. Results

### 3.1. Characterization

FI-TR spectra were collected to confirm the existence of related functional groups in materials (Figure 1). For the as-synthesized MTX-PEG-COOH, the peaks at 1720 cm^−1^ and 1108 cm^−1^ were attributed to the stretching vibration absorption of C=O and C-O-C, respectively. The IR band in the range of 3300–3700 cm^−1^ was assigned to methylene merging peak, and that at 2800–3000 cm^−1^ was assigned to the merging peak of hydroxyl and amino groups. The two peaks at 1290 and 1250 cm^−1^ were from the stretching vibration of the C-N group. In nHA, the bands at 1095 and 1036 cm^−1^ were ascribed to the stretching vibration peaks of P-O, and the bands at 606 and 563 cm^−1^ were caused by P-O bending vibration. Besides, the absorption in the range of 3300–3700 cm^−1^ was derived from the hydroxyl merging peak. Compared with the ^1^H NMR of NH_2_-PEG-COOH, that of MTX-PEG-nHA showed two amino peaks at 6.84 ppm and 6.61 ppm, several hydroxyl peaks at 7–8 ppm, and the methyl peak on the tertiary ammonia connected with the benzene ring on the methotrexate molecule at 3.21 ppm. These results indicated the successful synthesis of MTX-PEG-COOH. Compared with nHA, there was the stretching vibration absorption of C-N at 1230 and 1251 cm^−1^, as well as the stretching vibration absorption of methylene merging peak in the range of 2800–3000 cm^−1^ in the FI-TR spectra of MTX-PEG-nHA, which can be attributed to the successful preparation of MTX-PEG-nHA.

The TEM image proved that the prepared MTX-PEG-nHA exhibited a spindle irregular shape (Figure 2A). The ζ-potential of the nanocomposite was found to be –9.1 ± 0.5 mV (Figure 2B). The size of MTX-PEG-nHA measured by DLS was 100.4 ± 5.6 nm with a polymer dispersity index (PDI) of 0.197 ± 0.024.

### 3.2. Loading and Releasingof Drugs

Drug loading capacities of MTX-PEG-COOH and MTX-PEG-nHA monitored by UV-vis spectroscopy were found to be 21.3 ± 1.6% and 5.6 ± 0.23%, respectively. The results suggested that nHA had the capability to load MTX. The pH-responsive release profiles of MTX from MTX-PEG-nHA were studied in three different pH conditions. As presented in Figure 3, free MTX molecules were quickly released in physiological conditions, and the release rate reached to 87.52 ± 3.24%. The release rates of MTX-PEG-nHA within 48 h reached 39.14 ± 2.73%, 22.35 ± 1.14%, and 18.62 ± 0.64% at pH 5.0, 6.4, and 7.4, respectively, which were significantly lower than that of free MTX (*p* < 0.001) (Figure 4). The result demonstrated that MTX-PEG-nHA had a relatively slow and sustained drug release. Notably, the release rate at pH 5.0 was higher than that at pH 6.4 (*p* < 0.01) and at pH 7.4 (*p* < 0.001). The result indicated that MTX-PEG-nHA exhibited acid-responsive property, which can be ascribed to the unstable ester bond between the carboxyl group of PEG and the hydroxyl group of nHA, as well as the dissolution of nHA in the acidic conditions at physiological temperature [30,31]. Thus, MTX-PEG-nHA can be potentially used as a pH-responsive drug nanocarrier, and it can release MTX under the acidic environment at the tumor area and intracellular lysosomes.

### 3.3. Biocompatibility Assay

CCK-8 was used to examine the cytotoxicity of MTX and PEG-COOH. As shown in Figure 5, free MTX at the maximum concentration (*p* < 0.01) exhibited higher cytotoxicity toward 143B, HOS and MG63 cells in contrast to MTX-PEG-COOH with the same content of MTX. Meanwhile, the IC50 values of MTX-PEG-COOH in the above two kinds of cells are larger after multiplying by the MTX content (*p* < 0.01), which indicated that PEGylation has little effect on the cytotoxicity of MTX.

Subsequently, we tested the cytotoxicity of MTX-PEG-nHA and nHA to normal cells (HSFs) and three common osteosarcoma cells (143B, MG63 and HOS), and the cell viabilities measured at different concentrations after 48 h incubation were presented in Figure 5. As the concentrations increased, the cell viability of HSF still maintained above 90% even at the concentration of 40 μg/mL, demonstrating that MTX-PEG-nHA and nHA had little toxicity towards normal cells (*p* > 0.1 for the maximum dose group and control group). However, MTX-PEG-nHA showed a significantly enhanced inhibitory effect on three kinds of osteosarcoma cells, compared with nHA alone. The result indicated that the MTX-PEG-nHA targeted a series of osteosarcoma cells with different sensitivity [29]. At the maximum concentration, the cell survival rates of 143B, HOS and MG63 were 35.45 ± 2.44%, 43.77 ± 2.78%, and 65.05 ± 2.33%, respectively. The IC50 values were found to be 22.95 ± 1.15, 30.69 ± 1.57, and 81.78 ± 8.23 μg/mL (Table 1). The IC50 of MTX-PEG-nHA multiplied by the drug loading was lower than that of MTX for three osteosarcoma cells (*p* < 0.01), indicating that the nanoparticles could reduce the dosage of MTX [32]. As a supplement to CKK-8 assay, the morphology images of cells before and after treatment with MTX-PEG-nHA are illustrated in Figure 5B. Compared with the control, the addition of MTX-PEG-nHA showed an obvious effect on the cell morphology, indicating that almost all cells undergo apoptosis.

## 4. Discussion

Ligand-targeted nanoparticle-based drug delivery systems have been broadly studied in the treatment of various diseases. As a folate analogue, MTX can act as targeting ligand and enable the nanocarrier to bind selectively to folate receptor over-expressed on cancer cells, resulting in the increased drug delivery to the target cells. For example, Li et al. found that modification of MTX onto the surface of nano-diamonds could enhance the uptake of breast cancer cells. In the nano-delivery systems, MTX plays a dual-functional role: anticancer reagent and cell target. It can effectively target tumor cells while reducing drug dosage. Drugs were always covalently attached to the surface of nanocarriers. Insufficient intracellular drug release from nanoparticles will limit the amount of drugs that reach target area, ultimately decreasing the efficacy of chemotherapy [33]. In previous reports, MTX was released hardly from inorganic nanoparticles and thus reduced the cytotoxicity to tumor cells, although it could increase cell uptake. Through receptor-ligand-mediated endocytosis, pH-sensitive nanoparticles can be hydrolyzed, resulting in the release of drugs under acidic conditions in lysosomes. Therefore, drug delivery systems based on pH-sensitive bonds have been widely constructed for the therapy of diseases [10,30].

In this study, three different pH conditions (7.4, 6.4, 5.0) similar to the pH values of normal tissue, tumor microenvironment, and intracellular lysosomal microenvironment were employed to evaluate the acid responsiveness of nanoparticles. The results showed that the drug release was notably higher at lower pH value. Firstly, the formed ester bond between hydroxyl group on nHA and carboxyl group on PEG exhibited certain acid sensitivity. Secondly, nHA also exhibited a unique pH-dependent dissolution feature in the acidic cellular environments, including endosomes (pH 5.0) and lysosomes (pH 4.5) [34,35]. MTX released from nHA existed in the form of free status or as grafted with PEG. We have estimated the toxicities of grafted and non-grafted MTX, and found that PEGylation had little effect on its toxicity. Thus, MTX can be effectively released from nHA through the carrier self-destruction and further play a therapeutic role.

HA was commonly encapsulated with organic block copolymer for the applications of filling of bone defects [36,37]. For example, Zhang et al. fabricated nHA-loaded porous titanium scaffold to implant into a critical-sized segmental bone defect and found that nHA-releasing vehicle could inhibit tumor growth and accelerate bone regeneration [18]. Moreover, the accumulation of nano-hydroxyapatite in the bone area will stimulate the immune response. In this study, we integrated nHA into targeted drug delivery to increase cell uptake. The size of the as-prepared MTX-PEG-nHA was about 100.4 nm, which shows higher bioactivity toward the inhibition of the growth of osteosarcoma cells [16]. Moreover, nHA can more effectively extravasate into the tumor sites by the EPR effect [21]. The zeta potential (−9.1 ± 0.5 mV) provides adequate repelling force among MTX-PEG-nHA to form a stable system [38]. According to previous reports, the intracellular uptake and biodegradation of HA can lead to an increase in the concentration of calcium ions, and this further led to a decrease in the content of ATP due to mitochondrial membrane damage, thus increasing the sensitivity of chemotherapy drugs [19,20]. In this work, we combined nHA and MTX together to increase the intracellular uptake of nHA based on the characteristics of MTX. The obtained nanoparticles showed good efficacy in the treatment of three kinds of osteosarcoma cells in vitro, witha prolonged function time and a reduced dosage of MTX that exhibited low toxicity in normal cells. The introduction of nHA reduced the IC50 value calculated by MTX content in three kinds of cells. We also found that the sensitivity of the three osteosarcoma cells in relation to nanoparticles was different, which may be caused by the different expression of folate receptor or growth rate of the three osteosarcoma cells. Besides, the relationship between the degree of substitution of organic coating and inorganic core is not investigated systematically. With the increased amount of polymers, the water solubility and the drug loading rate may be further improved.

## 5. Conclusions

Herein, we developed a pH-responsive targeted drug delivery system for MTX using nHA as the biodegradable drug nanocarrier. MTX could target folate receptoroverexpressed on the osteosarcoma cells and improve the cell intake of nanocarriers. nHA exhibited pH-responsive biodegradable characteristic to release MTX at low pH (pH 5.0) and could reduce the dosage of MTX. In vitro drug release studies indicated the sustained release of MTX from the nanoparticles. MTX-PEG-nHA showed an enhanced effect on the cytotoxicity, with reduced dosage of MTX. The MTX-PEG-nHA is a promising candidate as a pH-responsive targeted drug delivery system for the chemotherapy and treatment of osteosarcoma.

## Data Availability

Not applicable.

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
