# Peer review of "Co-Delivery of Methotrexate and Nanohydroxyapatite with Polyethylene Glycol Polymers for Chemotherapy of Osteosarcoma"

_micromachines, 2023, doi:10.3390/mi14040757_

Round 1

Reviewer 1 Report

The methodology section throughout lacks references.

Line 116 and  123 : the amount of chemicals and reagents placed need to be optimized. The authors reported throughout the fixed amount/volume. Either quote specific reference on this or summarized the trials with results.

In the discussion section, the effect of zeta potential and particle size and its distribution is not discussed , although for the nanopaticulate system it decides the overall stability of the product

Drug release kinetics studies need to be required along with release data

How does in vitro drug release, simulate the in vivo conditions? Elaboration is required. 

Author Response

Comment 1: "The methodology section throughout lacks references."

Response: We have quoted the corresponding reference in the methodology section..

Comment 2: "Line 116 and  123: the amount of chemicals and reagents placed need to be optimized. The authors reported throughout the fixed amount/volume. Either quote specific reference on this or summarized the trials with results."

Response: We have quoted the corresponding reference in the methodology section.

Comment 3: "In the discussion section, the effect of zeta potential and particle size and its distribution is not discussed, although for the nanopaticulate system it decides the overall stability of the product."

Response: We have discussed the important role of the particle size and zeta potential in terms of targeting and treatment efficacy and stability of nanoparticles.

Comment 4:  "Drug release kinetics studies need to be required along with release data."

Response: We have provided the detailed release dates in PBS solution at different pH values.

Comment 5: "How does in vitro drug release, simulate the in vivo conditions? Elaboration is required. "

Response: It is a good question. We used PBS solution at three different pH conditions (7.4, 6.4, 5.0) to simulate the corresponding pH conditions of normal tissue, tumor microenvironment and intracellular lysosomal microenvironment.

Reviewer 2 Report

The manuscript by Ou et al. reports on a drug delivery system that is claimed to be capable of targeted delivery of methotrexate and nanohydroxyapatite for chemotherapy of osteosarcoma. Although the authors demonstrated the drug release and cytotoxicity behavior of the drug delivery system, the poor experimental design and insufficient data provided to support the conclusion lead me to believe that this paper does not meet the standards required for publication in Micromachines.

Major comments:

1.     How long is the PEG in this drug delivery system? The authors used a 3000 KDa dialysis tube to purify MTX-PEG-COOH, indicating that the PEG was quite lengthy (>3000k). Since methotrexate (MTX) is quite hydrophobic, if the PEG length is too long, it may encapsulate the MTX, thereby limiting the targeting ability of MTX. 

2.     The authors claim that the drug delivery system can target osteosarcoma, but they have not conducted any experiments to support this claim.

3.     The authors purified MTX-PEG-COOH using a dialysis tube against water. However, due to its amphiphilic nature, MTX-PEG-COOH may form micelles in aqueous solution, potentially resulting in the encapsulation of impurities such as free MTX within these micelles.

4.     The authors should include TEM and DLS data for MTX-PEG-nHA in the manuscript.

5.     How could methotrexate (MTX), which is insoluble in water, release so rapidly in the release experiment?

6.     The authors should consider using molar concentration instead of weight concentration in the cell viability study since MTX, MTX-PEG-COOH, and MTX-PEG-nHA have different molecular weights.

Author Response

We thank the reviewer for his/her comments: "The manuscript by Ou et al. reports on a drug delivery system that is claimed to be capable of targeted delivery of methotrexate and nanohydroxyapatite for chemotherapy of osteosarcoma. Although the authors demonstrated the drug release and cytotoxicity behavior of the drug delivery system, the poor experimental design and insufficient data provided to support the conclusion lead me to believe that this paper does not meet the standards required for publication in Micromachines." We have provided more detailed and clear exposition of the experimental procedures and discussion.

Comment 1: "How long is the PEG in this drug delivery system? The authors used a 3000 KDa dialysis tube to purify MTX-PEG-COOH, indicating that the PEG was quite lengthy (>3000k). Since methotrexate (MTX) is quite hydrophobic, if the PEG length is too long, it may encapsulate the MTX, thereby limiting the targeting ability of MTX."

Response: We are sorry for this mistake. The average molecular mass of PEG is 2000 Da. To investigate the in vitro drug release, a dialysis bag with molecular weight cut-off about 7000 Da was used, in which free MTX or MTX-PEG released from nHA was collected for subsequent calculation.

Comment 2: "The authors claim that the drug delivery system can target osteosarcoma, but they have not conducted any experiments to support this claim."

Response: MTX conjugated on the surface of nHA can specifically interact with folate receptor over-expressed in cell membranes of cancer cells, including osteosarcoma cells. In this work, we investigated the cytotoxicity of MTX-PEG-nHA toward normal cells and osteosarcoma cells. In contrast to normal cells, MTX-PEG-nHA showed a significantly enhanced inhibitory effect on osteosarcoma cells, indicating that they could target osteosarcoma cells.

Comment 3: "The authors purified MTX-PEG-COOH using a dialysis tube against water. However, due to its amphiphilic nature, MTX-PEG-COOH may form micelles in aqueous solution, potentially resulting in the encapsulation of impurities such as free MTX within these micelles."

Response: In this work, MTX was conjugated with a low molecular weight of PEG (2000 Da), which can decrease the encapsulation of free MTX.

Comment 4: "The authors should include TEM and DLS data for MTX-PEG-nHA in the manuscript."

Response: We are sorry for the mistake. Figure 2 shows the TEM and DLS data for MTX-PEG-nHA but not nHA.

Comment 5: "How could methotrexate (MTX), which is insoluble in water, release so rapidly in the release experiment?"

Answer: It’s a good question. According to the previous work reported by Baker’s group, the release rate of free MTX when dialyzed in PBS are more than 70% within 2.5 h (Patri, A. K.; Kukowska-Latallo, J. F.; Baker, J. R., Jr., Targeted drug delivery with dendrimers: Comparison of the release kinetics of covalently conjugated drug and non-covalent drug inclusion complex. Adv Drug Deliv Rev 2005, 57, 2203-2214).

Comment 6: "The authors should consider using molar concentration instead of weight concentration in the cell viability study since MTX, MTX-PEG-COOH, and MTX-PEG-nHA have different molecular weights."

Response: In this work, the drug loading capacity of MTX-PEG-COOH was calculated to be 21.3 ± 1.6% by UV-vis spectroscopy. Thus, the molar concentrations of MTX content in MTX-PEG-COOH with the corresponding weight concentrations were the same as that of free MTX during the cytotoxicity evaluation of free MTX and MTX-PEG-COOH. Meanwhile, the molecular weight of MTX-PEG-nHA was difficult to be exactly calculated and the use of weight concentrations facilitate readers to understand well. Therefore, the weight concentrations of MTX-PEG-nHA and nHA were used to evaluate the cytotoxicity of MTX-PEG-nHA and nHA toward different types of cells.

Round 2

Reviewer 2 Report

The authors thoroughly answered every question raised.